# Bladder Epicheck Test: A Novel Tool to Support Urothelial Carcinoma Diagnosis in Urine Samples

**DOI:** 10.3390/ijms241512489

**Published:** 2023-08-06

**Authors:** Vincenzo Fiorentino, Cristina Pizzimenti, Mariausilia Franchina, Esther Diana Rossi, Pietro Tralongo, Angela Carlino, Luigi Maria Larocca, Maurizio Martini, Guido Fadda, Francesco Pierconti

**Affiliations:** 1Department of Human Pathology of the Adult and Developmental Age “G. Barresi”, University of Messina, 98125 Messina, Italy; mariausilia.franchina@studenti.unime.it (M.F.); guido.fadda@unime.it (G.F.); 2PhD Programme in Translational Molecular Medicine and Surgery, Department of Biomedical and Dental Sciences and Morphofunctional Imaging, University of Messina, 98125 Messina, Italy; cristina.pizzimenti@unime.it; 3Department of Women, Children and Public Health Sciences, Catholic University of the Sacred Heart, Agostino Gemelli IRCCS University Hospital Foundation, 00168 Rome, Italy; esther.rossi@unicatt.it (E.D.R.); pietrotralongo@gmail.com (P.T.); angela.carlino@guest.policlinicogemelli.it (A.C.); francesco.pierconti@unicatt.it (F.P.); 4Department of Medicine and Surgery, Saint Camillus International University of Health and Medical Sciences (UniCamillus), 00131 Rome, Italy; luigimaria.larocca@unicamillus.org

**Keywords:** bladder carcinoma, urinary biomarkers, methylation, tumoral recurrence

## Abstract

Bladder cancer and upper urothelial tract carcinoma are common diseases with a high risk of recurrence, thus necessitating follow-up after initial treatment. The management of non-muscle invasive bladder carcinoma (NMIBC) after transurethral resection involves surveillance, intravesical therapy, and cytology with cystoscopy. Urinary cytology, cystoscopy, and radiological evaluation of the upper urinary tract are recommended during follow-up in the international urological guidelines. Cystoscopy is the standard examination for the first assessment and follow-up of NMIBC, and urine cytology is a widely used urinary test with high sensitivity for high-grade urothelial carcinoma (HGUC) and carcinoma in situ (CIS). In recent years, various urinary assays, including DNA methylation markers, have been used to detect bladder tumors. Among these, the Bladder EpiCheck test is one of the most widely used and is based on analysis of the methylation profile of urothelial cells to detect bladder neoplasms. This review assesses the importance of methylation analysis and the Bladder EpiCheck test as urinary biomarkers for diagnosing urothelial carcinomas in patients in follow-up for NMIBC, helping cytology and cystoscopy in doubtful cases. A combined approach of cytology and methylation analysis is suggested not only to diagnose HGUC, but also to predict clinical and histological recurrences.

## 1. Introduction

Urothelial carcinomas of the bladder and upper urothelial tract are widespread diseases related to exposure to several external environmental factors (such as tobacco and various chemicals) and chronic inflammation (for example, caused by persistent infections), and are characterized by a higher prevalence among Caucasian males. They carry a consistent risk of recurrence, thus implying a follow-up period after initial treatment. The use of urinary cytology, cystoscopy, and radiological evaluation of the upper urinary tract is recommended during follow-up by several international urological guidelines [1,2].

Cystoscopy is the standard examination for the first assessment and follow-up of non-muscle invasive bladder carcinoma (NMIBC) [3]. However, repeated cystoscopic examinations cause discomfort to patients and represent the principal factor in the high costs for the follow-up of NMIBC [4]. Therefore, urinary cytology has been employed as a complementary and more manageable tool for NMIBC follow-up, showing high sensitivity in high-grade urothelial carcinoma (HGUC) [5] and carcinoma in situ (CIS) but low sensitivity in low-grade urothelial carcinoma (LGUC). Moreover, the results of this test are observer-dependent [6]. Given the limited reliability of urinary cytology, many diagnostic urinary biomarkers have been studied over the last few years to identify recurrences while at the same time avoiding cystoscopic examinations. These are based on a deeper understanding of the molecular background of urothelial carcinomas. Such neoplasms are characterized by a variety of genetic alterations, and different molecular subtypes of urothelial carcinoma have been identified based on gene expression and genomic profiling [7].

Specifically, the Food and Drug Administration (FDA) and European Medicines Agency (EMA) approved bladder tumor antigen (BTA), nuclear matrix protein 22 (NMP22), ImmunoCyt/uCyt+, and UroVysion as urinary biomarkers for the detection and surveillance of urothelial carcinomas [8]. The BTA test is an antibody-based diagnostic tool that measures the urinary levels of the complement factor H-related protein (CFHrp), a molecule that is secreted by healthy cells to protect against autoimmunity, and its release has also been demonstrated in tumor cells as a potential aid in evading host immune defenses [9,10,11]. When compared to cytology, the BTA test has higher sensitivity but lower specificity because it can be influenced by several factors, such as inflammatory conditions, administration of Bacillus Calmette–Guerin (BCG), calculi, foreign bodies, previous instrumentation, bowel interposition, or concomitant genitourinary neoplasms, which can provide false-positive results [9,12]. Similarly, NMP22 is more common in neoplastic urothelial cells than in their non-malignant counterparts, and the process of apoptosis is associated with urine excretion of this protein, which is up to 25 times higher in bladder cancer [9,10,11]. Specifically, NMP22 belongs to the nuclear matrix protein (NMP) group, which is an essential component of the nuclear architecture that acts as a scaffold to provide structural integrity and shape to the nucleus. These proteins are involved in DNA replication, ribonucleic acid transcription, and gene expression regulation. However, NMP22 excretion can also occur due to urothelial cell apoptosis in non-malignant pathologies. Elevated NMP22 levels have been observed in individuals with urinary infections, concomitant urolithiasis, a history of bladder interposition, other neoplasms, intravesical therapies, and even cystoscopic examinations, similar to the BTA test. These factors may lead to false-positive results [9,13]. While the BTA and NMP22 tests are based on the detection of urinary molecules, the other tests detect cellular changes. The ImmunoCyt/uCyt+ test is an immunocytological fluorescence test that adopts a combination of three monoclonal antibodies to identify antigens exclusively derived from transitional epithelial cell tumors [9,10,11]. Two of these antibodies are conjugated with fluorescein (a green fluorescent dye) and bind to a mucin-like antigen found in the urine of exfoliated neoplastic cells, while the other is conjugated with Texas Red (a red fluorescent dye) and binds to a high-molecular-weight glycosylated variant of carcinoembryonic antigen. ImmunoCyt/uCyt+ is not a diagnostic tool in itself, but rather a monitoring test used in conjunction with cytology. Some authors have shown that ImmunoCyt/uCyt+ has higher sensitivity than cytology for pathological stage Ta-T2 and grade 1–2 tumors and comparable or increased sensitivity for grade 3 tumors and CIS [14,15]. However, other studies have reported a lower sensitivity [16]. In contrast to molecular diagnostic assays, ImmunoCyt/uCyt+ test results are unaffected by inflammatory conditions or instillation therapy. However, the observed variability in reported sensitivity may be attributed to factors such as observer expertise, and specimen stability, handling, and size [9]. In contrast, the UroVysion assay is a fluorescence-based cellular test that employs fluorescence in situ hybridization (FISH) to allow the direct visualization of chromosome copy numbers and specific DNA sequences within the cellular nucleus [9,10,11]. Chromosomal aberrations are common in the pathogenesis of bladder cancer (particularly on chromosomes 1, 3, 5, 7, 9, 11, and 17), with the loss of the 9p21 locus of chromosome 9, which houses the p16 tumor suppressor gene, being the most common and earliest genetic alteration observed. The UroVysion assay employs a multitargeted group of probes that bind to the centromeres of chromosomes 3, 7, and 17, as well as to the 9p21 locus. This assay has been shown to have a higher sensitivity than urinary cytology while maintaining comparable or marginally lower specificity [9,17]. Moreover, sensitivity was positively correlated with cancer grade across all studies. Similar to the ImmunoCyt/uCyt+ test, the results are unaffected by concurrent non-malignant conditions since the FISH technique analyzes nuclear alterations. The observed variability in the performance of the UroVysion assay can be attributed to several factors, including variations in scoring criteria, the use of voided urine versus bladder-wash urine, observer expertise, and specimen stability and handling. Furthermore, this method is burdened by non-negligible costs and necessitates the use of skilled technical personnel [9].

Globally, when compared to urine cytology, most of the above-mentioned urinary biomarkers show increased sensitivity with comparable or lower specificity. However, the heterogeneity of outcomes among studies is high [17], and some authors have highlighted the limitations of such screening tests over time [18]. Therefore, cystoscopy cannot currently be replaced by any of these biomarkers, and the need to search for new biomarkers is becoming increasingly important. For example, recent advancements in mRNA (e.g., “Xpert^®^ BC” [19]) and protein-based enzyme-linked immunosorbent assay (ELISA) analytical biochemical technique (e.g., “UBC^®^” [20]) have led to the development of several commercially accessible assays with enhanced sensitivity and specificity; however, their efficacy still needs to be verified in further independent research.

In this context, the Bladder EpiCheck Test, a novel urine assay that analyzes the degree of methylation of a group of genes implicated in bladder carcinogenesis, could represent an interesting alternative [21].

## 2. Bladder EpiCheck Test

DNA methylation is an epigenetic mechanism that alters gene expression without changing the underlying DNA sequence, and represents a well-known oncogenic process in several cancers, such as urothelial carcinomas [22]. In general, this process involves both tumor suppressor genes and oncogenes that when have a promoter hypermethylation or a hypomethylation, usually shows a low or iper expression. It has been postulated that this mechanism could be at the base of oncogenic processes and could represent a target for tailored therapies in an era in which tumor molecular signatures may represent crucial points in personalized medicine [23,24]. For example, in colorectal cancer, abnormal DNA hypermethylation of potential tumor suppressor genes has been identified as a promising biomarker for cancer screening, and it has recently been clinically applied [25]. Moreover, DNA methylation has been correlated with disease progression in NMIBC [21,26].

The Bladder EpiCheck test (Nucleix Ltd., Pekeris 3, Rehovot 7670203, Israel) analyzes 15 methylation biomarkers and detects the presence of urothelial neoplastic cells based on their methylation profiles. This test is performed on urine samples processed by centrifugation (twice at 1000× *g* for 10 min at room temperature) and DNA extraction from the cell pellet (using the Bladder EpiCheck DNA extraction kit). Then, the DNA is digested using a methylation-sensitive restriction enzyme, which cleaves the DNA at its recognition sequence if it is unmethylated. Subsequently, quantitative real-time polymerase chain reaction (qPCR) amplification is performed using a real-time platform and the results are analyzed using Bladder EpiCheck software version 1.9. The outcome of this assay is defined as the EpiScore; it consists of a number between 0 and 100, and a value ≥ 60 indicates a positive result [high risk for HGUC], and a score < 60 indicates a high probability of no bladder cancer or that the cancer is still in remission [negative or low risk for HGUC] [26]. In particular, an EpiScore of ≥90 indicates a diagnosis of HGUC. This test showed a higher sensitivity than urine cytology during the follow-up of NMIBC patients but a significantly lower specificity [27].

## 3. Evidence Acquisition

### 3.1. Search Strategies, Selection of Studies, and Data Extraction

Databases, including MEDLINE, Embase, and the Cochrane Library, were searched. Combinations of the following keywords were used to search for results up to March 2023: “non-muscle-invasive carcinoma”, “urinary biomarkers”, “urinary cytology”, “cystoscopy”, “Bladder EpiCheck test”, and “DNA methylation”. No language restrictions were applied. Moreover, reviews and articles on Bladder EpiCheck and upper urinary tract carcinoma were included. All studies, abstracts, and non-full text articles concerning the use of Bladder EpiCheck in patients with bladder carcinoma were included without exclusion criteria.

### 3.2. Studied Population

All cohort patients had a histologically proven diagnosis of NMIBC and were undergoing surveillance and therapy for NMIBC. Any stage (CIS, non-invasive papillary carcinoma (Ta), carcinoma with invasion of connective tissue (T1), without clinically identified regional lymph node metastasis (cN0), or clinically demonstrated metastasis (cM0)) according to the tumor, node, and metastasis (TNM) staging system (all versions) was included, and no gender limit was considered. In all studies considered, data from patients with a first diagnosis and patients who underwent intravesical therapy and surveillance cystoscopies at 3–6-month intervals were also included; data from patients with muscle-invasive bladder carcinoma (T2) and patients with non-localized disease [with neoplastic involvement of lymph nodes and clinically demonstrated distant metastases (cN1-3 and cM1)] were excluded.

Moreover, we included all studies on neoplasms of the upper urothelial tract with the same pathological and clinical criteria adopted for bladder carcinoma.

### 3.3. Types of Intervention and Outcome Measures Included

The bladder EpiCheck test was performed on patient urinary samples (voided, bladder washing, and ureteral washing) collected before standard cystoscopy. The sensitivity and specificity of Bladder EpiCheck were compared to those of cystoscopy, cytology, and histology results.

### 3.4. Data Analysis

Information on study design, inclusion and exclusion criteria, patient data, and outcome measures were analyzed in all consulted studies.

## 4. Evidence Synthesis

### 4.1. Characteristics of the Studies Included

Eleven studies that recruited 2516 patients met the inclusion criteria. All the studies were prospective, blinded, and single-cohort studies, with five regarding multicentric tumors [21,27,28,29,30] and six regarding monocentric tumors [26,31,32,33,34,35].

Two studies performed secondary external independent analyses with and without additional patients [27,30], and one study compared the sensitivity of two different urinary markers using qPCR in a population analyzed in a previous study [29].

### 4.2. Reference Standard Definition and Follow-Up

The sensitivity and specificity of Bladder EpiCheck were compared with a reference standard defined as the results of cystoscopy, cytology, and/or histology. The definition of the reference standard is critical for evaluating the results of Bladder EpiCheck in terms of positivity or negativity for this test. Histological examination was performed to better define cases with positive or suspected cystoscopy results for carcinoma recurrence, both on endoscopic biopsy and transurethral resection of the bladder (TURB). Mapping biopsies were planned for patients with positive or suspicious cytology and negative cystoscopy. Negative white-light cystoscopy, negative urinary cytology, and negative histology are the criteria for defining a patient as negative for recurrence of urothelial carcinoma. All studies used “The Paris System for Reporting Urinary Cytology” (TPS) for cytological specimen classification [36], and a negative urinary cytology was defined when the result was “Negative for High Grade Carcinoma” (NHGUC), while a diagnosis of “Atypical Urothelial Cells” (AUC) and “Suspicious for High Grade Carcinoma” (SHGUC) identified suspicious cases of recurrence of urothelial carcinoma. In cases with positive urinary cytology and negative cystoscopy and histology, a follow-up was performed with cystoscopy and multiple random biopsies or target biopsies when cystoscopy showed a suspicious area [32].

In the study by Pierconti et al. [32], a cytological diagnosis of AUC was considered positive, and patients underwent cystoscopy within three months with multiple random biopsies. In the study by Witjes et al. [21], a histological diagnosis of bladder carcinoma represented the only criterion for defining a recurrence of the neoplasm; in fact, in cases with histology negative for carcinoma and a cytological diagnosis positive for high-grade carcinoma, or in cases with positive or suspicious cystoscopy without a confirmatory biopsy, the sample was considered inconclusive and excluded from the analysis.

The follow-up of patients after bladder resection varies from 3 months [34] to a median follow-up of 3 years [33]. In the Witjes study, follow-up data from Bladder EpiCheck were not available [21].

### 4.3. Statistical Methods

All studies analyzed the sensitivity of Bladder EpiCheck to evaluate the performance of the test in detecting low- and high-grade urothelial carcinoma. A Receiver Operating Characteristic (ROC) curve and the area under the curve (AUC) were used to measure the EpiScore continuous variables. The positive predictive value (PPV), negative predictive value (NPV), and ROC curve were used to evaluate the accuracy of Bladder EpiCheck. To predict the presence of bladder carcinoma, D’Andrea et al. [27] generated two nomograms using the association between Bladder EpiCheck results and disease recurrence. The authors then used the Bladder EpiCheck results to make decisions in routine clinical practice.

## 5. Bladder EpiCheck Test and Bladder Carcinoma

### 5.1. Bladder EpiCheck Test and HGUC: Performance in Primary Diagnosis and in Diagnosis of Recurrence of Neoplasia

The first validation study of Bladder EpiCheck was performed by Wasserstrom et al. [31], who analyzed a population of 222 patients. In 40 of 222 patients, a histological diagnosis was performed according to the 2017 TNM classification and graded using both the 1973 and 2004 World Health Organization (WHO) classifications [37,38]. The Bladder EpiCheck sensitivity was higher in tumors with a high grade and stage (81%, 100%, 100%, and 91% in Ta, T1, T2, and CIS, respectively). In 2018, Witjes et al. [21] investigated the accuracy of Bladder EpiCheck in NMIBC and analyzed 353 of a 440-patient cohort in a multicenter, single-arm, prospective, and blinded cohort study, whose population underwent NMIBC surveillance with cystoscopy and cytology follow-up. The inclusion criteria were: (1) patients with recurrence of bladder urothelial carcinoma (UC) undergoing cystoscopic surveillance at 3-month intervals (adjuvant intravesical therapy allowed) or patients with a first diagnosis of bladder carcinoma; (2) all UCs resected within the previous 12 months; (3) all patients provided informed consent; and (4) age ≥ 22 years. Radical cystectomy, chemotherapy, and radiation for UC were considered exclusion criteria. They found that, excluding low-grade carcinoma, Bladder EpiCheck could be used in the diagnosis of bladder carcinoma, showing a relevant sensitivity with high specificity and high NPV (sensitivity, 91.7%; NPV, 99.3%; specificity, 88.0%). Based on these results, the authors suggested incorporating Bladder EpiCheck into NMIBC follow-up because high-grade recurrence would be instantly detected with a high likelihood. Moreover, they demonstrated that Bladder EpiCheck is an easily performable test that serves as a rule-out test and helps avoid unnecessary follow-up cystoscopies in NMIBC follow-up since it is possible to detect high-grade tumor recurrences with high confidence. Furthermore, they concluded that cytology with Bladder EpiCheck could reduce the significant burden of cystoscopy and cytology, thus reducing expenses for urologists and healthcare systems.

A subsequent study by D’Andrea et al. [27] confirmed this finding. In fact, these authors analyzed urinary samples from 440 patients in follow-up for NMIBC collected in five centers and studied the impact of Bladder EpiCheck in routine clinical practice to exclude patients with a high probability of recurrence. The inclusion criteria were a diagnosis of NMIBC < 12 months before entering the study and cystoscopy and cytology according to guideline recommendations to exclude recurrence of bladder carcinoma. The specificity for cancer detection was very high, and the NPV for the detection of any cancer (94.4%) reached 99% for the detection of HGUC.

In the multivariate analysis, the authors observed that positive Bladder EpiCheck results were independently associated with high-grade disease recurrence of bladder carcinoma, and that the implementation of Bladder EpiCheck into standard variables significantly improved its predictive ability for any high-grade disease recurrence. The data seem to suggest that Bladder EpiCheck as a diagnostic test in patients with NMIBC during follow-up can potentially reduce the number of unnecessary cystoscopies.

In 2019, Palacio et al., in a cohort of 657 patients, demonstrated an overall sensitivity of 62.5% for Bladder EpiCheck, which increased to 86% after excluding low-grade NMIBC, with a specificity of 86%. The NPV was 94% for all grades and 98% for non-low-grade carcinoma [30].

The increase in sensitivity of Bladder EpiCheck in HGUC was confirmed by Trenti et al. in two different studies, showing a sensitivity of 83% and 78% in NMIBC, with an overall sensitivity of 62% and 64%, respectively. The NPV considering all tumor grades was 83% and 89%, respectively [28,29]. The clinical data regarding Bladder EpiCheck performance are summarized in Table 1.

In a recent study, Pierconti et al. [26] analyzed 290 patients diagnosed with NMIBC who were followed up for 1 year after treatment using cytology, cystoscopy, and Bladder EpiCheck. In particular, intravesical BCG therapy was used in 216 patients, while mitomycin was used in 74 patients. From a histological perspective, the tumors were classified as high-grade papillary carcinoma G3T1 in 143 cases, G2T1 in 105 cases, and CIS in 42 patients. All cases were reviewed by different pathologists using the method described in previous papers by the same group [39,40]. 

In accordance with the European Association of Urology (EAU) Guidelines, all patients underwent white-light cystoscopy and voided urine cytology evaluations during the follow-up [1]. The urine specimens examined were prepared using the Papanicolaou staining method [41], and the cytological diagnosis was made in accordance with the TPS criteria.

The authors demonstrated that higher EpiScores were associated with a higher probability of identifying histological recurrence of HGUC, confirming the cytological diagnosis of HGUC or SHGUC, following the classification of the Paris System. More precisely, by analyzing the correlation between the histological diagnosis of high-grade carcinoma and EpiScore, they found that the methylation level increased progressively in concomitance with the rise in recurrence of high-grade urothelial carcinoma, from a value of 25% for EpiScore <70 to 90% in patients with EpiScore > 90. An EpiScore between 60 and 69 was observed in 25% of patients with recurrence of HGUC, between 70 and 79 in 64% of patients, and between 80 and 89 in 75%, while with an EpiScore > 90, the recurrence of HGUC was diagnosed in 90% of patients (these data are shown in Figure 1).

Moreover, the authors demonstrated that patients who had a cytologic diagnosis of HGUC or SHGUC with EpiScores ≥ 60 and negative histology for HGUC in 42% of cases showed HGUC recurrence, confirmed both cytologically and histologically, at 6–12 months during follow-up. This study allowed, for the first time, stratification of the risk of HGUC diagnosis in patients during follow-up using the EpiScores, thus validating Bladder EpiCheck not only in terms of the diagnosis of HGUC but also as an analytic tool for predicting HGUC recurrences during follow-up of NMIBC patients.

### 5.2. Bladder EpiCheck Test Combined with Urinary Cytology and Cystoscopy

Numerous studies have compared the sensitivity, PPV, and NPV for both cytology and Bladder EpiCheck, but very few have combined these two tests. All studies seemed to confirm that the sensitivity of Bladder EpiCheck was significantly higher than that of cytology, while its specificity could not reach a high value for cytology. In a study by Wasserstrom et al. [31], the sensitivity of cytology was 38%, with a specificity of 96%, while Bladder EpiCheck showed a sensitivity and specificity of 96% and 83%, respectively. Trenti et al. [29] showed that the better results of Bladder EpiCheck in terms of sensitivity were confirmed either by considering overall sensitivity (64.1% for Bladder EpiCheck vs. 27.2% for cytology) or by analyzing the sensitivity of low-grade or high-grade NMIBC, with values of 53.7% and 78.9%, respectively, while the sensitivity of cytology for low-grade carcinoma was 12.9%, with a value of 47.4% for high-grade carcinoma. The specificity of cytology was higher than that observed for Bladder EpiCheck (98.8% vs. 82.1%). The PPV and NPV for cytology were 86.2% and 83.6%, respectively, whereas those for bladder epithelium were 49.2% and 89.4%, respectively. Righetto et al. [33] showed an overall sensitivity of cytology and Bladder EpiCheck for the diagnosis of bladder carcinoma of 35.7% and 76.2%, respectively, and a specificity of 96.8% for cytology and 90.2% for Bladder EpiCheck. The sensitivity of Bladder EpiCheck to high-grade NMIBC was double that of low-grade carcinomas. The use of Bladder EpiCheck in combination with cytology was investigated by Trenti et al. [28]. This study included 243 patients, followed by voided urine cytology, the Bladder EpiCheck test, and white-light cystoscopy, according to the EAU guidelines. Photodynamic cystoscopy was performed in patients with positive cytology without evidence of a bladder tumor. Cystoscopically suspicious lesions were biopsied or removed by a transurethral procedure, and the specimens were evaluated according to the 2017 TNM classification of urinary bladder carcinoma and graded according to both the 1973 and 2004 World Health Organization classifications [37,38]. The results of this study showed an overall sensitivity of 33.3% for cytology, 62.3% for Bladder EpiCheck, and 66.7% for a combination of both tests. The combination of the two techniques reached an overall sensitivity of very low for low-grade (LG) bladder carcinoma (48.7%), while for high-grade (HG) bladder carcinoma, the sensitivity reached a value of 90%. The two tests combined showed a specificity of 85.6%, while the specificities for cytology and Bladder EpiCheck were 98.6% and 86.3%, respectively. The PPV was 92% for cytology and 68.2% for the bladder epithelium. For the two tests combined, it was 68.6%. The NPV was similar for the two tests: 75.8% for cytology, 82.9% for Bladder EpiCheck, and 84.5% for the combination of these techniques.

The combination of Bladder EpiCheck and cystoscopy with photodynamic diagnostics (PDD) has been studied by Cochetti et al. [42]. They determined the diagnostic performance of Bladder EpiCheck and PDD-guided cystoscopy in the surveillance of high-risk bladder cancer and compared the results with cytology. In this blinded, single-arm study, 40 patients under surveillance for high-risk NMIBC underwent cystoscopy with PDD and Bladder EpiCheck, setting those who received a histological diagnosis as a reference population. The inclusion criteria were age > 18 years, no bladder resection within 3 months, and urine cytology negative for infections. In these patients, bladder carcinoma was diagnosed 15 days prior to the visit. Muscle-invasive bladder carcinoma (MIBC) within 3 months, urinary infection, and recent therapy (chemotherapy or BCG intravesical instillation within <40 days) were considered exclusion criteria. All data, including the Bladder EpiCheck test, were collected and performed prior to PDD endoscopy, and all patients underwent urothelial carcinoma and cystoscopy with PDD in the operating room. For Bladder EpiCheck, the sensitivity and specificity of bladder cancer recurrence detection were very high, with values similar to those reported in the literature, whereas for PDD, the sensitivity and specificity were 61% and 41%, respectively. The data regarding the sensitivity and specificity of Bladder EpiCheck, cytology, and cystoscopy performance are summarized in Table 2.

Moreover, a recent paper analyzing Bladder EpiCheck and urinary cytology demonstrated that this molecular test seems to validate the new cytological categories introduced by the Paris System Classification of Urinary Cytology (TPS) [32]. Pierconti et al. studied 374 patients diagnosed with high-grade NMIBC who were treated and followed for 1 year with voided urine cytology, white-light cystoscopy, and biopsies. This cohort consisted of 268 patients with high-grade papillary carcinoma and 106 with CIS. After performing Bladder EpiCheck and cytology, they compared them with different cytological categories of the TPS: NHGUC was associated with an EpiScore <60 when compared to atypical urothelial cells (AUCs), while comparing the AUCs and SHGUC or SHGUC and HGUC, they found that an EpiScore ≥ 60 correlated with SHGUC and HGUC, respectively. They also pointed out that the sensitivity, specificity, PPV, and NPV of the Bladder EpiCheck test in the HGUC category were higher than those in the SHGUC group. This showed that different TPS cytological categories are linked to different molecular signatures and confirmed that SHGUC and HGUC should be considered as different entities (these data are shown in Table 3).

In recent years, several articles have shown that Bladder EpiCheck is a valuable tool for the diagnosis of HGUC, aiding pathologists in cases with equivocal urine cytology. Peña et al. [43] performed a study of 70 patients diagnosed with HGUC NMIBC. Forty percent of the cytological samples were catalogued as AUCs, while the DNA methylation test was positive in 17 urine samples, negative in 51, and in 2 samples, the results were not amplified. The authors demonstrated that the DNA methylation test can be used in the follow-up of patients with HGUC.

In patients with NMIBC at follow-up, the number of cytological diagnoses with an uncertain diagnosis of HGUC recurrence increases because of cytological alterations induced by therapy (BCG or Mitomycin C) [40,44,45]. In these cases, the analysis of methylation levels in urinary samples seems to allow for a correct diagnosis of HGUC.

Pierconti et al. [34] prospectively enrolled 151 patients with high-grade NMIBC. All the patients were treated with BCG and Mitomycin C intravesical therapy. Voided urine cytology and cystoscopy, with Bladder EpiCheck at the same time, were performed during the follow-up. Histology supported the diagnosis in every case in which the cytological results were positive. The specificity rates for Bladder EpiCheck and urine cytology were very similar at three months of follow-up (85.1% vs. 86.3%), as well as in the CIS group, while the specificity of Bladder EpiCheck was higher than that of cytology in patients with high-grade NMIBC with papillary histological morphology. The Bladder EpiCheck sensitivity was always higher than that of cytology during all follow-ups, both for high-grade papillary NMIBC and CIS. The ROC curve analysis of the Bladder EpiCheck test and cytology to predict a histological diagnosis of HGUC showed that at three months of follow-up, the diagnostic efficacy of Bladder EpiCheck was higher than that of cytology (these data are shown in Figure 2).

Another prospective single-center study was performed by Ragonese et al. [46] in a population of 231 patients during follow-up for NMIBC. The authors compared Bladder EpiCheck and urinary cytology and evaluated two endpoints: the evaluation of sensitivity and Bladder EpiCheck in detecting any type of bladder cancer recurrence. The secondary endpoint evaluated the specificity and sensitivity of Bladder EpiCheck both in patients with high-risk recurrence and in those recently treated with endovesical therapy. Cytology’s NPV was lower than that of Bladder EpiCheck’s (83 vs. 89%), as was its PPV (67% vs. 73% for cytology). In the high-grade NMIBC-only group, the sensitivity, specificity, and NPV of Bladder EpiCheck compared to urine cytology favored Bladder EpiCheck, with similar results in the cohort of patients with ongoing or recent endovesical treatment.

Hekman et al. [47] recently analyzed the cost of the Bladder EpiCheck follow-up strategy and demonstrated that including Bladder EpiCheck in the EAU Clinical Guidelines on NMIBC not only increased healthcare costs but also reduced them by 8% and 9%, respectively, in low- and high-risk patients. In fact, even though Bladder EpiCheck is more expensive than cystocopy (a difference of approximately EUR 30), follow-up with Bladder EpiCheck has shown a lower overall cost compared to the current strategy based on cytology and cystoscopy. This can be explained by the fact that Bladder EpiCheck reduces the number of cystoscopies by 40–32% and 42% in low-, intermediate-, and high-risk NMIBC, respectively, and reduces the number of negative TURBs by 24% in low- and intermediate-risk patients and 21% in high-risk patients in cases of false-positive cystoscopy or suspicious cytology.

Moreover, Pierconti et al. [48] have recently analyzed the combination of cytology and Bladder EpiCheck in follow-up of NMIBC patients, hypothesizing that it could represent an effective tool with a benefit for urologists, healthcare systems, and patients only in cases with a cytological diagnosis of AUC or SHGUC, while in cases with a cytological diagnosis of NHGUC or HGUC, cytology alone seems to be safe and cost-effective.

## 6. Bladder EpiCheck Test and Upper Tract Urothelial Carcinoma (UTUC)

This molecular test could be used to identify HGUC in all parts of the urinary tract and represents a valid tool in the diagnosis of UTUC.

A retrospective study by Pierconti et al. [35] analyzed 82 patients with high-grade UTUC (60 renal pelvis UTUCs and 22 ureteral UTUCs) who had undergone radical nephroureterectomy and had Bladder EpiCheck scores for urinary samples. The results correlated with those of urinary cytology and urethral biopsies. More precisely, before performing any surgical procedure, a urine sample obtained by selective ureteral catheterization for each patient was analyzed using cytology and Bladder EpiCheck. In a group of patients with HGUC, the results showed that the sensitivity of cytology for HGUC of the urothelial tract, considering only patients with a cytological diagnosis of HGUC, was 59% (from 47.25% to 69.99%; 95% CI (confidence interval)), with a specificity of 96% (from 86.29% to 99.51%; 95% CI). The sensitivity of cytology increased to 70.5% if we included patients with a cytological diagnosis of SHGUC. The Bladder EpiCheck results showed that the sensitivity of the methylation test for high-grade UTUC was 97.4% (from 91.04% to 99.69%; 95% CI) and the specificity was 100% (93.02 to 100%; 95% CI) (these data are shown in Table 4). The sensitivity of the latter test for high-grade UTUC was significantly higher than that of urinary cytology for both HGUC and SHGUC cytological diagnoses.

These results were confirmed by Territo et al. [49], who investigated the diagnostic accuracy of Bladder EpiCheck in the clinical management of UTUC by comparing it to urinary cytology. They performed a single-arm, blinded, prospective, single-center study with patients who were candidates for ureteroscopy for suspected UTUC. They collected bladder and upper urinary tract samples for cytology and Bladder EpiCheck, demonstrating that they were diagnostic in 97% of the upper urinary tract and bladder samples, while histological examination was positive in 57% and 58%, respectively. Histology was positive in 47/83 (57%) and 42/73 (58%) cases. EpiCheck’s statistical parameters in upper urinary tract samples were higher than those of cytology (sensitivity/specificity/NPV/PPV of 83%, 79%, 77%, 84%, 59%, 88%, 61%, and 87), as well as for high-grade tumors, with a few missed high-grade UTUCs and only 9% of the unnecessary ureteroscopies. This study demonstrated the potential value of EpiCheck in the assessment and management of UTUC, especially in the follow-up setting after conservative management. The use of EpiCheck in bladder or upper urinary tract urine should be carefully considered because the accuracy of the test should be balanced by its invasiveness.

## 7. Bladder EpiCheck Test and other Biomarkers in Urinary Cytology

In 2021, Wolfs et al. [50] evaluated the performance of urinary cytology and established biomarkers such as BTA, UroVysion, and ImmunoCyt, and critically reviewed the clinical efficacy of two new biomarkers (ADXBLADDER and Bladder EpiCheck). ADXBLADDER is an ELISA that detects minichromosome maintenance protein (MCM) 5 via MCM5 antibodies in the urine. MCM5 proteins are members of a family that plays a fundamental role in the initiation of DNA replication. They are present in proliferating cells, such as cancer cells, while their expression is low or absent in normal differentiated urothelial cells [51,52]. In three prospective studies, the sensitivity of ADXBLADDER (45–73%) and NPVs (74–100%) were superior to cytology, with acceptable specificity. These results indicated that this test could be a valuable tool for bladder carcinoma follow-up. The high NPV in patients undergoing follow-up implies that a negative result can be used to exclude patients with HGUC recurrence. Four Bladder EpiCheck studies reported that the overall sensitivity (62–90%) and NPV (79–97%) were superior to cytology, with a high specificity (82–88%). For the diagnosis of HGUC recurrence, the sensitivity and NPV of both molecular tests were high, while the specificity was lower than that observed in cytology during the follow-up of NMIBC. The ADXBLADDER and Bladder EpiCheck tests could potentially be used to exclude the presence of HGUC, and a possible reduction in cystoscopic investigations with an improvement in the quality of life of patients during follow-up for NMIBC is conceivable. In the future, these biomarkers may reduce the number of cystoscopies during follow-up, for instance, via a follow-up scheme that alternates between cystoscopy and biomarker tests. Moreover, the use of biomarkers in the follow-up of BC could represent an advantage in terms of reducing BC healthcare costs. The costs per cystoscopy in the United States are higher than those estimated for the ADXBLADDER and Bladder EpiCheck tests (approximately USD 225, 65, and 100, respectively).

The Xpert^®^ Bladder Cancer Monitor, which was studied by D’Elia et al. in 2019 [53], analyzes five target mRNAs (ABL1, CRH, IGF2, UPK1B, and ANXA10) in voided and stabilized urine using quantitative real-time reverse transcription PCR. More precisely, ABL1 is a protein-tyrosine kinase expressed in numerous neoplasias, and it has been detected in urine specimens of patients with bladder carcinoma, linked to the increased number of urothelial cells observed in these patients. UPK1B is a structural protein present in urothelial cells that is highly expressed in patients with bladder carcinoma. Moreover, CRH is a protein secreted from the hypothalamus and is involved in the regulation of stress responses and seemingly involved in the development of several tumors; ANXA10 is a calcium-dependent phospholipid-binding protein that plays a role in cellular growth. IGF2 mRNA is often upregulated in bladder carcinoma, and a high level of IGF2 protein is present in the urinary samples of bladder carcinoma patients. Based on these findings, CRH, ANXA10, and IGF2 were considered bladder carcinoma markers, and were therefore detected in urine samples. The Xpert^®^ Bladder Cancer Monitor detects target mRNAs using real-time RT-PCR, and the results are interpreted using a specific analysis system. The data reported by the authors seem to suggest that the sensitivity of the Xpert^®^ Bladder Cancer Monitor is higher than that observed in cytology in a large patient cohort, while the specificity seemed to not reach the high value of cytology. Moreover, the PPV and NPV were approximately the same for the Bladder EpiCheck and Xpert tests. Therefore, combining the Xpert^®^ test with voided urinary cytology could reduce cystoscopy in follow-up patients, with a reduction in discomfort and costs.

A recent study performed by Trenti et al. compared Bladder EpiCheck with a new biomarker test, the Xpert^®^ Bladder Cancer Monitor [29]. The authors enrolled 487 patients at the follow-up for NMIBC. During the follow-up, voided urine cytology, cystoscopy according to the current EAU guidelines, the Xpert^®^ Bladder Cancer test, and the Bladder EpiCheck test were performed. Cystoscopy was reserved for patients with positive cytology results and without visible bladder neoplasia. Cystoscopically suspicious lesions were biopsied or removed, and the specimens were diagnosed according to the 2017 TNM classification of urinary bladder cancer. All bladder carcinomas were graded according to both the 1973 and 2004 WHO classifications [37,38]. Patients without evidence of neoplasia after cystoscopy and negative cytology and histology were considered negative. For the Xpert^®^ Bladder Cancer test, the results were interpreted using the GeneXpert Instrument System, and the cut-off value was set at a laboratory-developed assay (LDA) of >0.5. This prospective study highlighted a greater overall sensitivity for Bladder EpiCheck (64.13%) and Xpert^®^ Bladder Cancer Monitor (66.3%) than for cytology (27.17%), while cytology showed a specificity higher than that observed with Bladder EpiCheck and Xpert^®^ Bladder Cancer Monitor (99%, 82%, and 76). When combined, Bladder EpiCheck and Xpert^®^ Bladder Cancer Monitor detected 79.35% of the tumors overall, 70.37% if we consider low-grade tumors, and 92.11% in patients with high-grade tumors, suggesting that the combination of these two tests could be advantageous to drastically reduce cystoscopy and cytology follow-up [53].

Laukhtina et al. [54] proceeded with a comprehensive review of several urinary biomarker tests for bladder cancer. Since none of these were recommended by international guidelines at that time, they investigated commercially available urinary assays for the diagnosis and surveillance of NMIBC, evaluating Xpert^®^ Bladder Cancer Monitor, Adxbladder, Bladder EpiCheck, Uromonitor, Cxbladder Monitor, and Triage and Detect. They found that in 21 published studies, the risk of bias was unclear. Novel urinary biomarkers showed sensitivities of up to 93%, specificities of up to 84%, PPVs of up to 67%, and NPVs of up to 99%. The number of potentially avoided cystoscopies proved that these tests may be efficient in reducing their necessity in many patients. The authors’ findings support the high diagnostic accuracy of the novel tests, confirming their utility in the NMIBC surveillance setting. However, there are insufficient data to reliably assess their use in a diagnostic setting.

Moreover, a recent review by Mancini et al. [55] compared Bladder EpiCheck and other markers used for follow-up of patients with NMIBC, showing that the sensitivity of the analysis of methylation patterns is higher than that observed in BTA (sensitivity 57–83%), ImmunoCyt/uCyt+ (sensitivity 50–100%), fluorescence in situ hybridization (UroVysion, sensitivity 70%), and NMP22 (sensitivity 47–100%). In the same paper, the authors highlighted the important disadvantages of Bladder EpiCheck: the cost of the test, the need for a molecular laboratory, and the need for a technician with experience in molecular analyses.

## 8. Conclusions

Bladder EpiCheck has demonstrated a high diagnostic performance in patients with NMIBC and can potentially reduce the number of unnecessary examinations. Moreover, numerous studies support the hypothesis that Bladder EpiCheck may represent a valid tool in the diagnostic process of HGUC patients, and for predicting recurrences, and could help in cases with difficult clinical decisions due to the technical limitations of cytology and histology. Nonetheless, the Bladder EpiCheck test may also benefit from the integration of new biomarkers to improve its performance. Rose et al. [56], for example, recently identified two promising DNA methylation biomarkers (*ITIH5* and *ECRG4*) for non-invasive identification of bladder carcinomas, which have the potential to be used in conjunction with the Bladder EpiCheck test to achieve better performance.

Overall, the Bladder EpiCheck test represents a valuable tool in the management of patients with bladder cancer owing to its high diagnostic accuracy, clinical utility, ability to reduce invasive diagnostic procedures, potential to improve biomarker panels, and cost-effectiveness.

However, further studies are needed to establish whether such an assay can be useful in the diagnostic and therapeutic strategies for NMIBC in non-Caucasian patients. In fact, it has been demonstrated that DNA methylation, as an epigenetic mechanism, plays a key role in the differences between human populations [57,58,59], and it represents a source of variability in human groups at macro- and micro-geographical scales. A strong genetic component and several factors, such as nutrients, UVA exposure, and different pathogens can influence DNA methylation profiles in the human population [60,61].

## Figures and Tables

**Figure 1 ijms-24-12489-f001:**
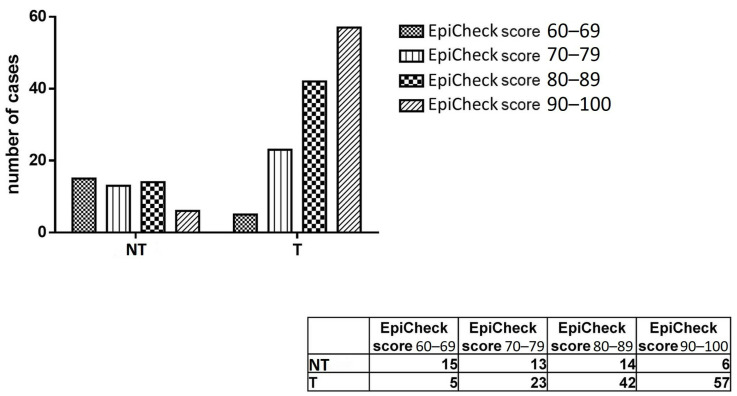
Correlation between the histological recurrences of high-grade urothelial carcinoma and the EpiScore values. On the x-axis and in table: NT and T on the x-axis and in the correlated table indicate no tumor and tumor, respectively. Reprinted/Adapted with permission from ref. [26]. Copyright year: 2022; Copyright Owner’s Name: Cancer Cytopathology.

**Figure 2 ijms-24-12489-f002:**
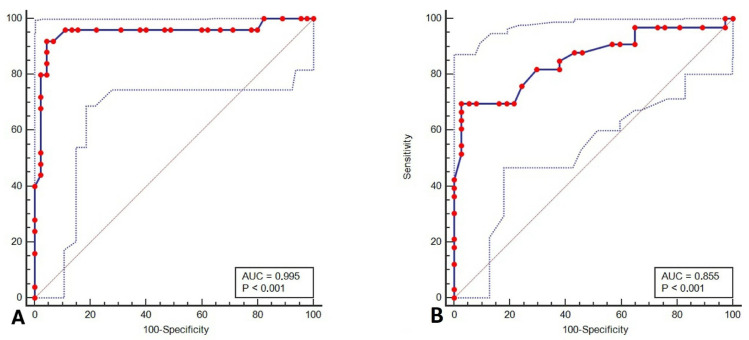
The ROC curves of Bladder EpiCheck (**A**) and cytology (**B**). In (**A**) the area under the curve (AUC) for the diagnostic accuracy was 0.995 (*p* < 0.001; 95% CI (confidence interval) from 0.870 to 0.988; Youden index J = 0.8756; sensitivity 92%; specificity 95,5%), while in (**B**) it was 0.855 (*p* < 0.001; 95% CI from 0.751 to 0.928; Youden index J = 0.6699; sensitivity 69.7%; specificity 97.3%). Reprinted/Adapted with permission from ref. [34]. Copyright year: 2022. Copyright Owner’s Name: Urologic Oncology: Seminars and Original Investigations.

**Table 1 ijms-24-12489-t001:** Performance of Bladder EpiCheck test (BE: Bladder EpiCheck test; PPV: positive predictive value; NPV: negative predictive value; LG: low-grade bladder carcinoma).

BE Number of Patients	Sensitivity All Grades (%)	Sensitivity Non-LG (%)	Specificity (%)	PPV (%)	NPV All Grades (%)	NPV Non-LG (%)	Ref.
222	90	95%	83	/	97	/	[31]
353	68.2	88.9	88	44.8	95.1	99.3	[21]
657	62.5	86.4	85.8	/	94.3	98.8	[30]
357	67.3	89	88	47	94	99.3	[27]
215	62.3	83.3	86.3	8.2	82.9	/	[28]
432	64.1	78.9	82.1	49.1	89.4	/	[29]

**Table 2 ijms-24-12489-t002:** Data regarding the sensitivity and specificity of different diagnostic tools for bladder cancer detection.

Cytology	EpiCheck	EpiCheck + Cytology	PDD Cystoscopy	Ref.
				[31]
38%	90%	/	/	Sensitivity
96%	83%	/	/	Specificity
				[29]
27.20%	64.10%	/	/	Sensitivity
12.90%	53.70%	/	/	- Low-grade NMIBC
47.40%	78.90%	/	/	- High-grade NMIBC
98.80%	82.10%	/	/	Specificity
86.20%	49.20%	/	/	PPV
83.60%	89.40%	/	/	NPV
				[33]
35.70%	76.20%	/	/	Sensitivity
0%	37.50%	/	/	- Low-grade NMIBC
50%	100%	/	/	- High-grade NMIBC
96.80%	90.20%	/	/	Specificity
				[28]
33.30%	62.30%	66.70%	/	Sensitivity
7.70%	46.10%	48.70%	/	- Low-grade NMIBC
66.70%	83.30%	90%	/	- High-grade NMIBC
98.60%	86.30%	85.60%	/	Specificity
92.00%	68.20%	68.60%	/	PPV
75.80%	78.60%	84.50%	/	NPV
				[42]
88.90%	100%	/	61.10%	Sensitivity
100%	90.90%	/	40.90%	Specificity
100%	90%	/	45.80%	PPV
91.70%	100%	/	56.30%	NPV

**Table 3 ijms-24-12489-t003:** EpiCheck score in different TPS categories (OR: odds ratio; CI: confidence interval).

	NHGUC	AUC	
EpiScore < 60	143	45	*p* = 0.0003 OR 3.925 95% CI from 1.907 to 8.081
EpiScore > or = 60	17	21	
	**AUC**	**SHGUC**	
EpiScore < 60	45	13	*p* = 0.0031 OR 3.791 95% CI from 1.612 to 8.915
EpiScore > or = 60	21	23	
	**SHGUC**	**HGUC**	
EpiScore < 60	13	14	*p* = 0.0027 OR 3.957 95% CI from 1.639 to 9.550
EpiScore > or = 60	23	98	
	**SHGUC**	**HGUC**	
EpiScore 60–89	22	52	*p* < 0.0001 OR 19.4 95% CI from 2.522 to 150.2
EpiScore > or = 90	1	46	

Reprinted/adapted with permission from ref. [32]. Copyright year: 2021; Copyright Owner’s Name: Journal of Clinical Pathology.

**Table 4 ijms-24-12489-t004:** Sensitivity, specificity, PPV and NPV of cytology and Bladder EpiCheck Test for UTUC.

	Cytology		Bladder EpiCheck Test	
Statistics	Value	95% CI	Value	95% CI
Sensitivity	58.97%	47.25% to 69.99%	97.44%	91.04% to 99.69%
Specificity	96.00%	86.29%to 99.51%	100.00%	93.02% to 100.00%
Disease Prevalence	60.94%	51.92% to 69.44%	60.47%	51.48% to 68.96%
PPV	95.83%	85.38% to 98.91%	100.00%	98.05% to 100%
NPV	60.00%	5333% to 66.32%	96.23%	86.65% to 99.01%
Accuracy	73.44%	64.91% to 80.85%	98.45%	94.51% to 99.81%

Reprinted/adapted with permission from the author [35]. Copyright year: 2021 Copyright Owner’s Name: Human Pathology.

## Data Availability

Data reported in this manuscript are a re-analysis of existing data and are available within the paper. All consulted studies are listed in the “References” section.

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
