# Peer review of "Bladder Epicheck Test: A Novel Tool to Support Urothelial Carcinoma Diagnosis in Urine Samples"

_ijms, 2023, doi:10.3390/ijms241512489_

Round 1
Reviewer 1 Report (New Reviewer)
I agree with the importance of methylation analysis and the Bladder Epicheck test as urinary biomarkers for diagnosing urothelial carcinomas in patients in follow-up for NMIBC, helping cytology and cystoscopy in doubtful cases. In this way we will can see a reduction in unnecessary access to the OR.
Author Response
Thanks to reviewer for the enthusiastic revision.
Reviewer 2 Report (New Reviewer)
Fiorentino et al. have submitted the review article entitled “Bladder Epicheck test: a novel tool to support the urothelial carcinoma diagnosis in urine samples” This review encompasses a brief overview of FDA approved markers in the urine for surveillance of NMIBC. The majority of the review focuses on Bladder EpiCheck and how it compares to cytology, cystoscopy and histology. The authors included relevant studies and provided summarization tables contrasting sensitivity/specificity/NPV/PPV.
Some comments to improve the manuscript:
-Line 379 -might soften wording regarding PDD use in setting of Epicheck; PDD provides opportunity to localize disease (if present) vs a non-visualized detection approach; would be dependent on clinical context
-Table 2 – need to adjust alignment on first row relative to represented %
Spelling/Grammar
– line 169 (cM0)
-line 179 – switch to bladder washing
-mixing of full word vs abbreviations (e.g., NPV, EAU) throughout text
Line 374- needs rewording
Line 412 – change to Forty percent
Author Response
Fiorentino et al. have submitted the review article entitled “Bladder Epicheck test: a novel tool to support the urothelial carcinoma diagnosis in urine samples” This review encompasses a brief overview of FDA approved markers in the urine for surveillance of NMIBC. The majority of the review focuses on Bladder EpiCheck and how it compares to cytology, cystoscopy and histology. The authors included relevant studies and provided summarization tables contrasting sensitivity/specificity/NPV/PPV.
Some comments to improve the manuscript:
-Line 379 -might soften wording regarding PDD use in setting of Epicheck; PDD provides opportunity to localize disease (if present) vs a non-visualized detection approach; would be dependent on clinical context
Following the reviewer’s suggestion, we modified the text, erasing an entire phrase (“Although Bladder EpiCheck has been demonstrated to be a valid diagnostic tool for the early identification of recurrences, the use of PDD in such patients should be reconsidered.”). The message is now less strong than before and more pertinent to the clinical context.
-Table 2 – need to adjust alignment on first row relative to represented %
The alignment in the table is now corrected in the new version of the manuscript.
Spelling/Grammar
– line 169 (cM0)
The specific error is now corrected in the new version of the manuscript.
-line 179 – switch to bladder washing
The mistake is now corrected.
-mixing of full word vs abbreviations (e.g., NPV, EAU) throughout text
Line 374- needs rewording
As the reviewer suggested, the indicated phrase in now rewording to make it clearer.
Line 412 – change to Forty percent
The mistake is now corrected.
Reviewer 3 Report (New Reviewer)
Authors proposed a paper entitled “Bladder Epicheck test: a novel tool to support the urothelial carcinoma diagnosis in urine samples” for the publication in International Journal of Molecular Sciences, MDPI.
The paper has a quite good scientific soundness, and deserves to be published after some revisions.
A list of acronyms should be added according to the guidelines of this journal: NMIBC,
HGUC, LGUC, BTA, FISH,
Line 44. “in several international urological” I would say “by”
Line 95. “reported sensitivity may be attributed to factors such as observer expertise” please add a reference here.
Line 97. “in situ” should be written in italics.
Line 131. “genes that are hypermethylated and oncogenes that are” check syntax here.
Line 132. “Several authors” authors say several authors, but only show 2 references. Please be consistent.
Line 149. “carcinoma (HGUC] ), and” remove additional space before the closing parenthesis
Line 106. “the results were included” and Line 161 “the results were included u”: authors should uniform their decision to use impersonal or personal forms.
Line 184. “For all papers” better saying that these papers were consulted.
Table 1. Reference could be reported in a final column with square parenthesis; this in order to save space for the table presentation.
Figure 1. I suggest giving a name to x axis. The table can be transofrmed into caption text or it can be moved out of the diagram figure and considered as table 2.
Table 2. Same observation of actual table 1 for references.
Check also the format of table 2, according to the guidelines of this journal.
Table 3 has a different format than the others. Check the guidelines.
A quite good use of english
Author Response
Authors proposed a paper entitled “Bladder Epicheck test: a novel tool to support the urothelial carcinoma diagnosis in urine samples” for the publication in International Journal of Molecular Sciences, MDPI.
The paper has a quite good scientific soundness, and deserves to be published after some revisions.
A list of acronyms should be added according to the guidelines of this journal: NMIBC,HGUC, LGUC, BTA, FISH.
Following the reviewer’s suggestion, a “List of Abbreviations” is now added after the conclusion section.
Line 44. “in several international urological” I would say “by”
The error is now corrected.
Line 95. “reported sensitivity may be attributed to factors such as observer expertise” please add a reference here.
As suggested, we added the specific reference.
Line 97. “in situ” should be written in italics.
The term in situ is now corrected in in-situ (italics) in the manuscript.
Line 131. “genes that are hypermethylated and oncogenes that are” check syntax here.
In the new version of the manuscript, the indicated phrase is changed to make it clearer.
Line 132. “Several authors” authors say several authors, but only show 2 references. Please be consistent.
The error is now corrected, following the reviewer’s suggestion.
Line 149. “carcinoma (HGUC] ), and” remove additional space before the closing parenthesis
The space before the closing parenthesis is now erased.
Line 106. “the results were included” and Line 161 “the results were included u”: authors should uniform their decision to use impersonal or personal forms.
Following the reviewer’s suggestion, we use impersonal form in the new version of the manuscript.
Line 184. “For all papers” better saying that these papers were consulted.
In the new version of the manuscript, the indicated phrase is changed to make it clearer.
Table 1. Reference could be reported in a final column with square parenthesis; this in order to save space for the table presentation.
In the new version of the manuscript, the Table 1 is modified following the reviewer’s suggestion.
Figure 1. I suggest giving a name to x axis. The table can be transformed into caption text or it can be moved out of the diagram figure and considered as table 2.
As reported, the Figure 1 is a partially reprinted figure, with permission, from our previously published scientific report (ref. 26). We decided to leave the figure as like the original as possible. Following the reviewer’s suggestion, we better explain what T and NT indicate on the x axis and in the table.
Table 2. Same observation of actual table 1 for references.
Check also the format of table 2, according to the guidelines of this journal.
In the new version of the manuscript, the Table 2 is modified following the reviewer’s suggestion.
Table 3 has a different format than the others. Check the guidelines.
As suggested, we modified the Table 3 to make it similar to others.
Round 2
Reviewer 3 Report (New Reviewer)
Authors proposed a new version of their paper.
The work has been revised according to reviewer’s issues.
Authors responded point by point to the issue required.
Table 1 needs to be revised accordingly. In particular, I suggest changing “studies” (last column) with “Refs”. Moreover, in order to save space, I would just write the numbers of references such as [X], [Y] instead of writing also names.
Same observation for Table 2.
Check Nr. “46” at last line of Table 3.
There is an empty cell in Table 4.
Line 494. “from ref. [35].” Please say “from Author [35]”
Author Response
Authors proposed a new version of their paper.
The work has been revised according to reviewer’s issues.
Authors responded point by point to the issue required.
Table 1 needs to be revised accordingly. In particular, I suggest changing “studies” (last column) with “Refs”. Moreover, in order to save space, I would just write the numbers of references such as [X], [Y] instead of writing also names.
We revised the table following the reviewer’s suggestion.
Same observation for Table 2.
We revised the table following the reviewer’s suggestion.
Check Nr. “46” at last line of Table 3.
The number 46 is now located in right space in the Table 3.
There is an empty cell in Table 4.
The value in now reported in the empty cell.
Line 494. “from ref. [35].” Please say “from Author [35]”
As the reviewer suggested, we modified the phrase.
This manuscript is a resubmission of an earlier submission. The following is a list of the peer review reports and author responses from that submission.
Round 1
Reviewer 1 Report
Pierconti et al. They present a manuscript on an interesting topic that may have significance. However, this manuscript is inadequately written and in maximum disorganization.
The authors must make an introduction to the state of the art and the natural history of the disease from a translational perspective and that is accessible to a specific public. In its current format it is simply an approximation of disclosure on bladder cancer. The methylation aspects are important, of course, but in this review it is not clear what focus and purpose the authors want and should give to this manuscript.
The quality of the writing is frankly deficient, with very serious errors from the scientific and grammatical point of view.
This reviewer does not understand the structure of this review, what is the purpose of section 4? and the 4.3?
What is the meaning of figure 1? whose is it? the quality is poor and nothing is understood.
The conclusions are intentional. I don't understand anything about this manuscript. This is not a narrative review that may be of interest in any area.
English very difficult to understand/incomprehensible
Reviewer 2 Report
The manuscript was well written and it is very original and interesting. Three main remarks: i) the abstract is too long and it should be shortened highlighting only the main information on the manuscript; ii) in the Introduction more information should be added about the main diagnostic urinary biomarkers already developed (authors only cited these markers but I think it would be better to explain them); iii) the conclusions should be written avoiding redundant information already discussed in the other sections and it should be focused only on the main results in relation to the aims of the study.Author Response
Please see the attachment.

Round 2
Reviewer 1 Report
The authors present the revised version of the manuscript "Bladder EpiCheck test: a novel biomarker for methylation study of urine samples". The authors have not made changes of any kind, they have only made small adjustments of no consequence. This manuscript is of low quality and lacks adequate interest. The quality of all the content is already known and there are documents of greater interest. This is a simple disclosure document.
English very difficult to understand/incomprehensible